# Role of pH and Crosslinking Ions on Cell Viability and Metabolic Activity in Alginate–Gelatin 3D Prints

**DOI:** 10.3390/gels9110853

**Published:** 2023-10-27

**Authors:** Andrea Souza, Matthew Parnell, Brian J. Rodriguez, Emmanuel G. Reynaud

**Affiliations:** 1School of Biomolecular and Biomedical Science, University College Dublin, D04 V1W8 Dublin, Ireland; andrea.souza@ucdconnect.ie (A.S.); matthew.parnell@ucdconnect.ie (M.P.); 2School of Physics, University College Dublin, D04 V1W8 Dublin, Ireland; brian.rodriguez@ucd.ie

**Keywords:** 3D printing, alginate–gelatin hydrogel, pH, CaCl_2_, BaCl_2_, U2OS, NIH/3T3, fluid phase

## Abstract

Alginate–gelatin hydrogels are extensively used in bioengineering. However, despite different formulations being used to grow different cell types in vitro, their pH and its effect, together with the crosslinking ions of these formulations, are still infrequently assessed. In this work, we study how these elements can affect hydrogel stability and printability and influence cell viability and metabolism on the resulting 3D prints. Our results show that both the buffer pH and crosslinking ion (Ca^2+^ or Ba^2+^) influence the swelling and degradation rates of prints. Moreover, buffer pH influenced the printability of hydrogel in the air but did not when printed directly in a fluid-phase CaCl_2_ or BaCl_2_ crosslinking bath. In addition, both U2OS and NIH/3T3 cells showed greater cell metabolic activity on one-layer prints crosslinked with Ca^2+^. In addition, Ba^2+^ increased the cell death of NIH/3T3 cells while having no effect on U2OS cell viability. The pH of the buffer also had an important impact on the cell behavior. U2OS cells showed a 2.25-fold cell metabolism increase on one-layer prints prepared at pH 8.0 in comparison to those prepared at pH 5.5, whereas NIH/3T3 cells showed greater metabolism on one-layer prints with pH 7.0. Finally, we observed a difference in the cell arrangement of U2OS cells growing on prints prepared from hydrogels with an acidic buffer in comparison to cells growing on those prepared using a neutral or basic buffer. These results show that both pH and the crosslinking ion influence hydrogel strength and cell behavior.

## 1. Introduction

Recently, 3D (bio)printing technology has shown its potential to replace and complement existing methods in basic research, drug delivery, screening, and medical procedures. For instance, it allows the production of a bioprinted full-skin equivalent, which can support bone and capillary formation and breast fat tissue growth, among other biological structures created in (bio)printed scaffolds in vitro [1,2,3,4]. However, even though some important advances have been made to date, this technology is still evolving and requires more research on the basic requirements needed to create optimal matrices to grow tissues in vitro. To construct a successful environment, the biological, physical, mechanical, and chemical aspects of these scaffolds should be established specifically for each tissue and/or purpose of the study, as pathologic tissues behave differently from healthy ones.

Hydrogels are largely used in the bioengineering field due to their ability to mimic the extracellular environment. Alginate hydrogel is the second most used natural bioink in this field because of its biocompatibility, biomimetic, tunability, good printability, and crosslink characteristics [5]. However, alginate’s highly swollen polymer crosslinked network can lead to instability and degradation in just a few days under cell culture conditions [6]. Therefore, tailoring the alginate into a more biomimetic matrix can make it a more attractive biomaterial. The addition of gelatin can improve the mechanical properties of the hydrogel, increasing its elasticity and decreasing its stiffness [7,8]. Moreover, Alsberg et al. (2001) have shown that the incorporation of RGD molecules into alginate gel transforms the hydrogel into a suitable matrix to promote bone formation [9]. Finally, an additional way to tailor its mechanical properties is the degree of crosslinking, which can be determined by the crosslinker type, concentration, temperature, time of exposure, etc. [10,11].

Alginate can be chemically or physically crosslinked. Chemical crosslinking involves the formation of irreversible covalent bonds between alginate chains and leads, in general, to better stability under cell culture conditions [12,13,14,15]. Physical crosslinking involves hydrogen bonding, hydrophobic bonding, ionic bonding, electrostatic interactions, etc., to create reversible three-dimensional structures [13,14,15,16]. Alginate is usually ionically crosslinked in the bioengineering field using divalent cations, including calcium. However, Ca^2+^ molecules from the crosslinked hydrogel can be released into the medium, causing inflammatory responses in vitro and in vivo [17]. Another possibility for the ionic crosslinking of alginate substrates for cell-based therapies is the use of Ba^2+^ molecules. Barium has a better affinity to alginate, therefore providing hydrogels with stronger mechanical properties and increased Young’s modulus in comparison to calcium [18]. Yet, Ba^2+^ has shown diverse effects on cell viability and proliferation depending on the cell type. For instance, Luca et al. (2007) reported that encapsulated Sertoli cells in alginate microbeads showed greater viability after barium crosslinking in comparison to calcium [19]. However, Mores et al. (2015) showed a decrease in the mononuclear phagocyte viability caused by apoptosis due to barium crosslinking [20]. Therefore, as few studies regarding barium crosslinking are available in the literature, alginate crosslinking with barium should be studied carefully for each cell type and condition.

Another important aspect during the preparation of hydrogel scaffolds for bioengineering is the pH of the solution. FitzSimons et al. (2022) described how pH influences the polymer bonding kinetics, the mechanical properties, and the protein release of PEG hydrogels crosslinked via reversible thia-Michael addition [21]. In addition, pH is involved in regulating the solubility of alginate/gelatin in the aqueous phase [22]. Bouhadir et al. (2001) showed that increasing the pH leads to an increase in the alginate hydrogel’s degradability [23]. In addition, the alginate hydrogel viscosity increased with decreasing pH, reaching a maximum viscosity at pH 3.0–3.5 once the carboxylate groups were present in the alginate chain protonate and formed hydrogen bonds [15]. Recently, the effect of pH on hydrogels has been largely studied, especially regarding the production of responsive hydrogels used to deliver drugs or optimize cell growth and differentiation. Tailoring hydrogel’s mechanical properties, degradation rate, and affinity to proteins via pH changes is a costless and important tool for the bioengineering field. Therefore, understanding the effect of hydrogel’s intrinsic pH as well as the surrounding pH on the hydrogel’s mechanics and kinetics is fundamental to accurately control hydrogel behavior.

In this work, we used a hydrogel composed of 6% alginate and 2% gelatin (*w*/*v*) to investigate the impact of different pH values (5.5, 6.5, 7.0, 8.0) and crosslinking ions (CaCl_2_, BaCl_2_) on hydrogel stability under cell culture conditions incubated with either DMEM or RPMI 1640. These two media are widely used in cell culture and present differences in their composition regarding calcium (RPMI: 0.8 mM, DMEM: 1.8 mM) and phosphate (RPMI: 5 mM, DMEM: 1 mM) concentrations [24]. Both media present a physiological pH of 7.0–7.4. The printability of the alginate–gelatin hydrogel with different pH values was tested in the air and under fluid phase using a support bath containing either 100 mM CaCl_2_ or 100 mM BaCl_2_. Air printing allowed the printability of high-viscosity biomaterials and bioinks, while fluid-phase printing allowed the printability of low-viscosity biomaterials and bioinks [25]. Finally, to broaden our understanding, different cell types (the human osteosarcoma cell line U2OS and the murine fibroblast cell line NIH/3T3) were used to study the effect of the substrate’s pH (as modified by buffer) and crosslinking on cell viability and metabolism.

## 2. Results and Discussion

### 2.1. Results

Alginate–gelatin hydrogels were prepared using an MES buffer with a pH of 5.5, 6.5, 7.0, or 8.0 and crosslinked with either 100 mM CaCl_2_ or 100 mM BaCl_2_ to study the role of the buffer pH and crosslinking on hydrogel stability over time. With all other factors remaining the same, the addition of a different pH buffer was expected to change the pH of the hydrogel, hereafter the substrate’s pH. Hydrogels were molded into 10 mm discs and kept under cell culture conditions incubated with either complete RMPI 1640 + 10% FBS or complete DMEM + 10% FBS for 25 days to measure their swelling and degradation rates. As a result, we observed that all the conditions presented a weight loss of approximately 50% during overnight crosslinking with either calcium or barium. However, they regained their initial weight at different speeds. Hydrogels crosslinked with calcium and incubated with RPMI were the least stable under cell culture conditions, showing, in general, the fastest degradation rate among all the conditions tested in this work. They regained their initial weight within 24 h, and the swelling of all the hydrogels was between 150 and 200% before degrading (Figure 1A). Crosslinking with calcium and incubation with DMEM showed better stability in comparison to the previous conditions. The swelling of the samples was around 135% before degradation. However, all the substrates, regardless of their pH, degraded within 25 days under cell culture conditions (Figure 1B). Crosslinking with barium provided overall good stability to the alginate–gelatin hydrogel. Incubation with RPMI 1640 promoted the higher swelling of samples (~150%) in comparison to their incubation with DMEM (~120%) (Figure 1C,D). Regarding the hydrogel pH, the hydrogel pH 8.0 presented the weakest stability under cell culture conditions in RPMI regardless of the crosslinking and in DMEM after calcium crosslinking (Figure 1).

Afterward, alginate–gelatin hydrogels with different pH values had their printability tested under the following three different conditions: air printing, fluid-phase embedding printing with 100 mM of a CaCl_2_ support bath, and fluid-phase embedding printing with 100 mM of a BaCl_2_ support bath. We reported in previous findings that to produce printings of the alginate–gelatin hydrogel, the fluid phase is an efficient way to increase printing fidelity and resolution [26]. To generate these prints, a needle size of 0.43 mm ID was used, and the diameter of the strand spread was measured just after printing for the three conditions. First, we observed that the spread of the air printing strand was around 200%, while both fluid-phase embedding printing restrained the spread of the printed strand as the hydrogel was immediately crosslinked during printing. In addition, the two crosslinker solutions were used as support baths and showed no significant difference in the spread of the strand between them (Figure 2A). The pH of the alginate–gelatin hydrogel had a significant influence on air printing, increasing the spread of the printed strand with the increase in pH. By contrast, fluid-phase embedding printings only showed a slight increase in the spread of the strand with a hydrogel pH of 8.0, which was not significant (Figure 2B). The spread of the strand is important for the resolution of the (bio)print and should be calculated and adjusted to obtain the desired printing design. Knowing how the pH of the substrate can influence this aspect and how to overcome it is an important parameter for the biofabrication field. Finally, our results showed that both crosslinker (Ca^2+^ and Ba^2+^)-containing support baths increased the printing fidelity, which is particularly important at the corners of the prints, as seen in Figure 2C.

Next, we tested the effect of crosslinking on the cell viability and metabolic activity of the human osteosarcoma U2OS and the murine fibroblast NIH/3T3. Both cell types are well established in the literature and present high-cell proliferative rates, making them good models for this study. U2OS cells did not show significant a difference in the cell death rate of cells growing on alginate–gelatin substrates crosslinked with calcium or barium either on day 1 or day 7 of cell culture (Figure 3A). However, the substrate crosslinked with barium led to a 3.62-fold decrease in U2OS cell metabolic activity in comparison to the substrate crosslinked with calcium (Figure 3B). NIH/3T3 cells presented a high-cell death rate on both day 1 and day 7 of cell culture on alginate–gelatin hydrogel crosslinked with barium at 45% and 60%, respectively, in comparison to 10% and 20%, respectively, of the cell death rate on the same hydrogel crosslinked with calcium (Figure 3C). Moreover, NIH/3T3 cells also showed less metabolic activity on the substrate crosslinked with barium (83.91%) in comparison to the one crosslinked with calcium (100%) (Figure 3D). Even though crosslinking with barium showed better stability of the alginate–gelatin hydrogel, the two cell types studied in this work did not show good cell viability and/or cell metabolic activity on substrates crosslinked with Ba^2+^.

To study the effect of the substrate’s pH on cell viability and metabolic activity, U2OS and NIH/3T3 were seeded onto alginate–gelatin prepared with the MES buffer of pH 5.5, 6.5, 7.0, and 8.0. Cells were kept in culture for 7 days incubated with complete RPMI +10% FBS (U2OS) or complete DMEM + 10% FBS (NHI/3T3). Our results show that U2OS cells growing on alginate–gelatin hydrogels prepared with different pH values present good cell viability of approximately 90% of viable cells on day 1 and day 7 of cell culture. Some cell protective effects were observed by the substrate at pH 7.0; however, no significant difference in cell viability was caused by the substrate pH (Figure 4A). On the contrary, the substrate presenting different pH values led to a significant difference in cell metabolic activity. Cells growing on substrates prepared with pH 7.0 and 8.0 showed an important increase in cell metabolism in comparison to cells growing on acidic substrates after 7 days of cell culture. An increase of 2.25-fold was observed in cell metabolism on the substrate at pH 8.0 in comparison to cells on the substrate at pH 5.5. All the cells were grown in an RPMI 1640 cell culture medium under physiological pH. The results show that it is possible to grow cells over the influence of the pH of interest by preparing a substrate with a specific pH independent of the cell culture medium utilized. In the same way, NIH/3T3 cells also showed good viability of cells regardless of substrate pH values on day 1 and day 7 of cell culture (Figure 4C). However, in the same fashion as U2OS cells, NIH/3T3 also showed a significant difference in cell metabolism due to the substrate pH. NIH/3T3 cells growing on substrate at pH 7.0 presented higher metabolic activity in comparison to cells growing on substrates at pH 6.5 and 8.0 and even greater cell metabolic activity in comparison to cells growing on substrates at pH 5.5 (Figure 4D).

Finally, we observed that the pH of the substrate also influenced the cell–cell and cell–matrix interactions, with cells favoring interactions with each other below pH 7 and the matrix at pH 7 and above. Alginate–gelatin substrates at pH 5.5 and 6.5 led to the formation of cell aggregates, whereas substrates at pH 7.0 and 8.0 showed cells that grew in a more spread-out layout after 7 days of cell culture (Figure 5).

### 2.2. Discussion

Alginate hydrogels can be crosslinked with many divalent cations. In the present study, we tested the swelling and degradation rates of alginate–gelatin hydrogels with different pH values after crosslinking with either 100 mM CaCl_2_ or 100 mM BaCl_2_. Both cations are widely used in the biofabrication field to crosslink the alginate for different purposes, such as drug encapsulation and delivery, wound dressing, tissue formation, etc. In addition, the choice of the crosslinker and its concentration can tailor some of the hydrogel’s mechanical properties like elasticity, strength, stiffness, swelling, etc. [27]. Haper and Barbut (2014) showed that alginate films, when crosslinked with BaCl_2_, had the highest tensile strength and Young’s modulus among Ba^2+^, Ca^2+^, Mg^2+^, Sr^2+^, and Zn^2+^ while films crosslinked with CaCl_2_ had the highest puncture strength [18]. Alginate’s affinity for divalent cations was shown to decrease in the following order: Pb^2+^ > Cu^2+^ > Cd^2+^ > Ba^2+^ > Sr^2+^ > Ca^2+^ > Co^2+^, Ni^2+^, Zn^2+^ > Mn^2+^ [28,29]. Gel strength decreased with decreasing affinity. Our study is in accordance with the literature, as barium crosslinking provided better stability for all the tested hydrogels in comparison to calcium crosslinking. Additionally, we observed a shrinkage of approximately 50% on all hydrogels incubated with both crosslinker solutions (CaCl_2_ and BaCl_2_) for 24 h. Saitoh et al. described a similar effect of alginate hydrogel when crosslinked with Ca^2+^ and incubated with increasing concentrations of CaCl_2_, which showed an increasing shrinkage rate. Increasing the binding between Ca^2+^ and alginate residual carboxylate groups led to an increase in the crosslinking degree, which facilitated gel shrinkage [30]. In the present study, we show that Ba^2+^ presents a similar effect to Ca^2+^ regarding the alginate hydrogel shrinkage extent after overnight incubation. In addition to the crosslinker effect on the hydrogel, we also showed that the cell culture medium in which the hydrogel was kept played an important role as hydrogels incubated with complete DMEM + 10% FBS presented overall better stability over time in comparison to the same hydrogels incubated with complete RPMI 1640 + 10% FBS, even though both media had the same physiological pH. In addition, among the hydrogels presenting different pH values, the hydrogel at pH 8.0 showed the weakest strength under cell culture conditions in general, which degraded faster than the hydrogels at pH 5.5, 6.5, and 7.0. Anionic hydrogels such as alginate swell at a high pH and shrank at a lower pH. The deprotonation of carboxylic groups of the alginate molecules at high pH decreased the strength of the hydrogel as the negatively charged ions repelled each other, leading to hydrogel swelling and fast degradation. In the opposite fashion, acidic media can lead to the protonation of alginate carboxylic groups, decreasing repulsion and causing shrinkage because of water loss [31,32,33]. In this study, we showed that we can also tailor the alginate hydrogel swelling and degradation/dissociation rates by changing the pH of the solvent used to produce the hydrogel independently of the medium’s pH.

This pH effect was also seen in the printability of the hydrogel, as the spread of the strand significantly increased with the increase in the hydrogel’s pH during printing in the air. However, fluid-phase embedding printing using either 100 mM CaCl_2_ or 100 mM BaCl_2_ as support baths was shown to be efficient in preventing spread from occurring. The slight increase in the spreading of the line of hydrogel at pH 8.0 printed in the fluid phase was not significant, as the immediate crosslink after printing the hydrogel was sufficient to keep its shape. No significant difference was seen between calcium and barium in relation to the printability of the hydrogel with different pH values. The barium support bath showed a lower correlation between the spread of the strand and the increasing pH of the hydrogel in comparison to calcium. Jui-Jung et al. (2017) described that barium crosslinking baths with different pH values do not influence the shape of alginate particles; however, crosslinking with calcium at lower pH values does exert an influence [34].

However, even though the crosslinking with barium provided more stability and strength to the alginate–gelatin hydrogel, both U2OS and NIH/3T3 cells showed significantly higher cell metabolic activity on hydrogels crosslinked with calcium in comparison to hydrogels crosslinked with barium. It was described that both crosslinkers, Ca^2+^ and Ba^2+^, presented a rate of release of molecules to the medium due to their relatively weak ionic interaction and competition with other cations present in the medium. Chan and Mooney (2013) described how alginate crosslinked with calcium releases around 43% of the Ca^2+^ incorporated in its meshes within the first 10 h of incubation with a cell culture medium and keeps releasing Ca^2+^ molecules at a lower rate thereafter as Ca^2+^ is slowly exchanged by sodium cations present in the cell culture medium [17]. Ba^2+^ also presented a high release rate from alginate in vitro and in vivo, which might be a safety concern [35]. In addition, it is widely known that Ca^2+^ is one of the most important intracellular second messengers participating in an extensive number of cell signaling pathways in relation to cell adhesion, proliferation, metabolism, apoptosis, etc. [36]. There is the possibility that Ba^2+^ ions released from alginate can compete with Ca^2+^ in important metabolic pathways, decreasing the metabolic rate of cells seeded onto the substrate crosslinked with Ba^2+^ [37,38,39]. However, there are few studies on the effect of barium crosslinking on cell activity and metabolism, and further investigations should be conducted for better understanding. We also observed that Ba^2+^ crosslinking did not increase cell death in comparison to Ca^2+^ crosslinking on U2OS cells, whereas Ba^2+^ crosslinking increased the NIH/3T3 cell death rate significantly.

To study the influence of substrate pH on cell viability and metabolism, U2OS and NIH/3T3 cells were seeded onto substrates with different pH values. Bone cells responded to even slight differences in the environment’s pH with higher osteoclast activity in acidic pH, while osteoblasts have higher activity in basic pH [40,41,42]. This is one of the homeostatic mechanisms to keep the systemic acid-base balance. Matsubara et al. (2013) showed that U2OS cells modulate their proliferation rate in response to extracellular pH, increasing proliferation in higher pH values [43]. In this work, we used the same cell type to investigate if changing the substrate pH could change cell behavior. Our results showed that a different substrate pH led to a significant difference in cell metabolism and no influence on cell viability. Corroborating the literature, osteoblasts growing on basic substrates showed greater cell metabolic activity in comparison to cells growing on acidic substrates. Our results, however, show that independently of the cell culture medium pH, the substrate’s pH played a crucial role in cell behavior. In addition, we observed that the U2OS cells can modulate the cell arrangement in response to the substrate pH, forming aggregates on acidic printings and spreading out on basic printings. Cells start to grow in 3D aggregates when the environment is unfavorable to cell–substrate interactions, leading cells to perform cell–cell and cell-ECM interactions instead [44].

Studying the interstitial pH of each tissue is a difficult task. There is no available data for human bone tissue. However, mice embryos were reported to have no impairment in their development in pH between 7.17 and 7.37 [45]. In this work, we used NHI/3T3 cells, which are murine embryonic fibroblasts. It is not reported from which tissue it originated, but Dastagir et al. (2014) showed that this cell type can differentiate into adipogenic, chondrogenic, and osteogenic lineages [46]. Our results showed that NHI/3T3 cells had a higher metabolic activity on substrates prepared with a buffer of pH 7.0 in comparison to substrates prepared with pH 5.5, 6.5, and 8.0, even though all cells were kept in a cell culture medium under physiological pH. In addition, the pH of the buffer did not influence NHI/3T3 cell viability. Future work in this direction should attempt to directly measure substrate pH as a function of time.

## 3. Conclusions

The bioengineering field is a very complex area of science because it involves several steps, which are all very specific to each cell type and circumstance. In this study, we aimed to understand the importance of carefully studying the pH of 3D prints. In addition, we also aimed to understand how crosslinking affects not only the mechanical properties of hydrogels but also influences cell viability, growth, and behavior on the prints. In essence, our findings show that the pH of the microenvironment in which the cells are in direct contact to grow influences cell behavior independently of the pH of the cell culture medium in the range studied. Moreover, barium crosslinking provided better stability for alginate–gelatin hydrogels independently of the substrate pH. However, U2OS and NIH/3T3 cells showed significantly lower metabolic activity when grown on substrates crosslinked with Ba^2+^ in comparison to Ca^2+^ crosslinking. In addition, Ba^2+^ crosslinking increased NIH/3T3 cell death. Therefore, Ca^2+^ has been shown to be better at crosslinking for the growth of both U2OS and NIH/3T3 cells on alginate–gelatin hydrogels. This illustrates that each cell type can respond differently to different substrate pH values and crosslinking protocols, indicating the importance of adjusting the conditions in 3D printing to achieve desired results.

## 4. Materials and Methods

### 4.1. Materials

Alginic acid sodium salt type 1, sodium chloride, calcium chloride, and EDC (1-ethyl-3-(3-dimethylaminopropyl) carbodiimide hydrochloride) were purchased from Thermo Fisher Scientific, Dublin, Ireland; for gelatin type B, bovine skin and Triton X 100 were purchased from Sigma-Aldrich Ireland Limited, Arklow, Co. Wicklow, Ireland; the MES buffer at pH 5.5, 6.5, 7.0, and 8.0, Polyetherimide (PEI), Barium Chloride, and NHS (N-hydroxysuccinimide) were purchased from Alfa Aesar, Ward Hill, MA, USA; MVG GRGDSP (RGD) was purchased from Novatech, Hattiesburg, MS, USA; RPMI 1640, DMEM, fetal bovine serum (FBS), and trypsin/EDTA were purchased from Gibco Life Technologies at Thermo Fisher Scientific, Dublin, Ireland. The U2OS human osteosarcoma cell line (ATCC^®^ HTB-96^TM^) and NIH/3T3 murine fibroblast cell line (ATCC^®^ CRL-1658^TM^) were used as cell models and purchased from ATCC, Manassas, VA, USA.

### 4.2. Methods

#### 4.2.1. Cell Culture

U2OS cells and NIH/3T3 cells were cultured in 75 cm^2^ flasks in complete RPMI 1640 and DMEM, respectively, both supplemented with 10% FBS under a saturated humidified atmosphere at 37 °C and 5% CO_2_. Subconfluent cultures were passaged at a ratio of 1:5 (U2OS cells) or 1:3 (NIH/3T3 cells) using a 0.05% trypsin/EDTA solution. High-density cells, 0.5 × 10^6^/cm^2^, were seeded onto alginate–gelatin printings coated with 120 µM/mL of RGD + 5% EDC + 2.5% NHS was diluted in 100 mM CaCl_2_ for 1 h [47,48].

Cells growing on alginate–gelatin hydrogel printings were monitored under an inverted phase contrast microscope (Olympus CKX53, Olympus Corporation, Tokyo, Japan), and photomicrographs were taken by phase contrast using the 10× objective. Images were analyzed by the software ImageJ/FIJI 2.9.0 (https://imagej.net/software/fiji/ [26], accessed on 12 September 2023).

#### 4.2.2. Alginate–Gelatin Hydrogel Preparation and Crosslinking

Adapted from [49], 2% gelatin (*w*/*v*) was added to a sterile 0.1 M MES buffer at pH 5.5, 6.5, 7.0 or 8.0 + 0.3 M NaCl at 50 °C and stirred for 10 min. Sodium alginate was then added at a final concentration of 6% (*w*/*v*) and stirred thoroughly at 50 °C until the hydrogel was homogeneous. Different hydrogel pH values were obtained with the use of an MES buffer with different pH values as a solvent. The hydrogel pH was not measured throughout the experiments. Hydrogels were either molded or printed in the air or fluid phase. Hydrogels were then crosslinked overnight with either 5 mL of 100 mM calcium chloride or 5 mL of 100 mM barium chloride, both dissolved in deionized water.

#### 4.2.3. Swelling and Degradation Measurement

In total, 6% alginate + 2% gelatin pH 5.5, 6.5, 7.0, or 8.0 were molded into 10 mm diameter discs and crosslinked with 5 mL of either 100 mM calcium chloride or 100 mM barium chloride. Hydrogels were weighed before crosslinking (W_0_) as a control. To measure the swelling and degradation rates, hydrogels were also weighed after crosslinking (W_ACl_), and on day 1 (W_D1_), day 5 (W_D5_), day 10 (W_D10_), day 15 (W_D15_), day 20 (W_D20_) and day 25 (W_D25_) under cell culture conditions, they were incubated with either complete RPMI 1640 or complete DMEM both supplemented with 10% FBS at 37 °C with saturating humidity. Hydrogels were air-dried for 30 min before having their weight measured. Calculations were made for the percentage of W_0_, and experiments were finished when 2/3 of the samples disintegrated.

#### 4.2.4. 3D Printing and Printing Resolution

The extrusion-based NAIAD 3D bioprinter, developed by the 3D Bioprinting Group from the Conway Institute at the University College Dublin, was used to print the alginate–gelatin hydrogel in the air and fluid phase. Briefly, hydrogels were warmed at 37 °C for 30 min in the bioprinter before the printing process. Alginate–gelatin printings were obtained using a 0.43 mm ID nozzle gauge 23 under 3 bars of pressure and 1mm/s of speed in the air or fluid phase containing either 100 mM CaCl_2_ or 100 mM BaCl_2_. Prior to the fluid-phase printing, six-well plates were treated with 0.1% PEI overnight for hydrogel attachment purposes and were washed 3 times with PBS to remove excess PEI. The width of the strands was measured immediately after printing and was analyzed via the software ImageJ/FIJI 2.9.0 (https://imagej.net/software/fiji/ [26], accessed on 12 September 2023).

#### 4.2.5. Cell Viability Assay

An LDH Cytotoxicity Detection Kit plus (Roche Diagnostics, Basel, Switzerland) was used to quantify U2OS and NIH/3T3 cell death on day 1 and day 7 of cell culture. Briefly, cells were incubated for 24 h with either RPMI 1640 or DMEM, each supplemented with 1% FBS, centrifuged, and the amount of LDH on the cell medium was measured at 490 nm by spectrophotometer (SpectraMax M3). Positive and negative controls were conducted on conventional 2D cell culture on plastic plates. The positive control was treated with 2% Triton X-100. Calculations were given as a percentage of the control.

#### 4.2.6. Adapted alamarBlue^TM^ Assay

To quantify the metabolic activity of U2OS and NIH/3T3 cells, 10% of the alamarBlue^TM^ cell viability reagent (Invitrogen) was added to the DMEM + 10% FBS (NHI/3T3) or RPMI 1640 + 10% FBS (U2OS) media on day 7 of cell culture and incubated for 4 hours. Next, the cell media containing reduced alamarBlue^TM^ were centrifuged, and 100 µL was transferred to a 96-well plate. The absorbance of the reduced alamarBlue^TM^ was measured at 570 nm and 600 nm. Calculations were performed according to the manufacturer’s instructions and given as a percentage of the control. For calculation purposes, the metabolic activity of cells growing alginate–gelatin hydrogel at pH 7.0 crosslinked with 100 mM of CaCl_2_ was considered a control (100%).

#### 4.2.7. Statistical Analysis

All statistical data processing was performed using multiple comparison tests on either One-way ANOVA or Two-way ANOVA on GraphPad Prism 9 software. Differences between the groups were considered reliable if *p* < 0.05 and values were expressed as the mean ± SD of at least 3 independent experiments.

## Figures and Tables

**Figure 1 gels-09-00853-f001:**
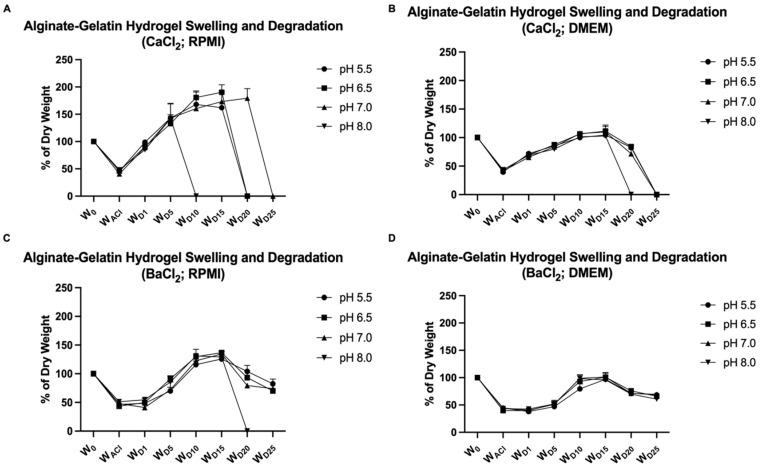
Alginate–gelatin hydrogel swelling and degradation curve. 6% alginate + 2% gelatin hydrogel prepared with 0.1 M MES buffer pH 5.5, 6.5, 7.0 or 8.0 and crosslinked with either 100 mM CaCl_2_ or 100 mM BaCl_2_. The samples were kept under cell culture conditions at 37 °C with saturating humidity with either RPMI 1640 + 10% FBS or DMEM + 10% FBS. Hydrogels were dried and weighed after being molded into 10 mm diameter discs (W_0_ = 100%), after crosslinking (W_Acl_), and on day 1 (W_D1_), day 5 (W_D5_), day 10 (W_D10_), day 15 (W_D15_), day 20 (W_D20_), and day 25 (W_D25_). (**A**) Hydrogels crosslinked with 100 mM CaCl_2_ and incubated with RPMI 1640 + 10% FBS. (**B**) Hydrogels crosslinked with 100 mM CaCl_2_ and incubated with DMEM + 10% FBS. (**C**) Hydrogels crosslinked with 100 mM BaCl_2_ and incubated with RPMI 1640 + 10% FBS. (**D**) Hydrogels crosslinked with 100 mM BaCl_2_ and incubated with DMEM + 10% FBS. (n = 4, graph of mean ± SD).

**Figure 2 gels-09-00853-f002:**
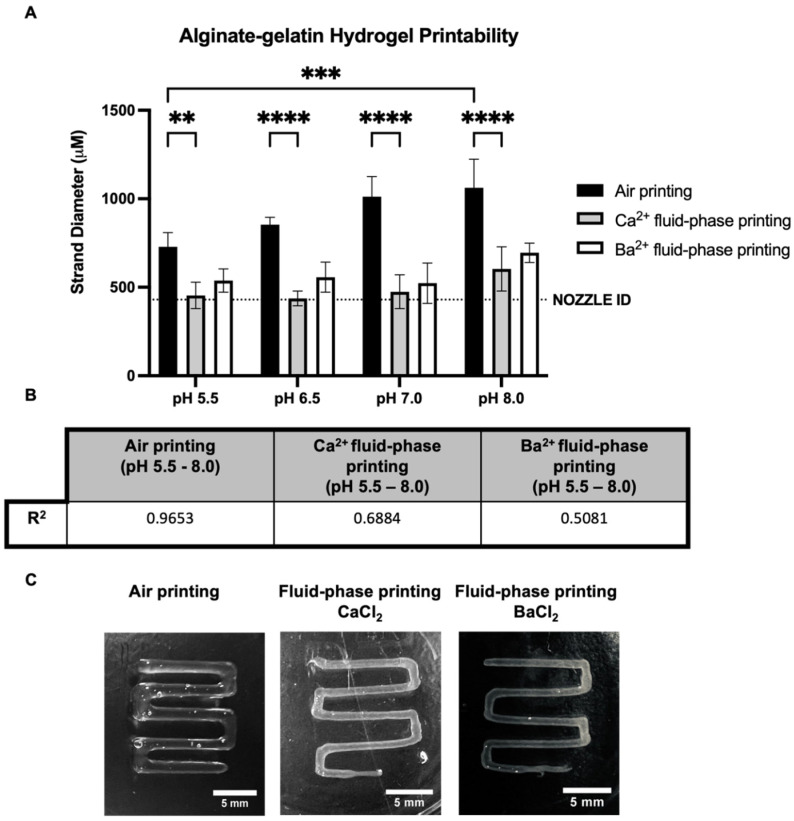
Alginate–gelatin hydrogel printability. (**A**) Quantification of the alginate–gelatin hydrogel at pH 5.5, 6.5, 7.0, or 8.0 for the diameter of the strand printed in the air and the fluid-phase embedding support bath containing either 100 mM CaCl_2_ or 100 mM BaCl_2_. (n = 4, Two-way ANOVA, ** *p* < 0.01, *** *p* < 0.001, **** *p* < 0.0001, ns: not significant). (**B**) R-squared of the effect of increasing pH of alginate–gelatin hydrogel on the spread of the printed line (strand). (**C**) Alginate–gelatin hydrogel pH 7.0 square wave printing obtained under 3 bars and 1 mm/s in the air and fluid-phase embedding support bath containing either 100 mM CaCl_2_ or 100 mM BaCl_2_.

**Figure 3 gels-09-00853-f003:**
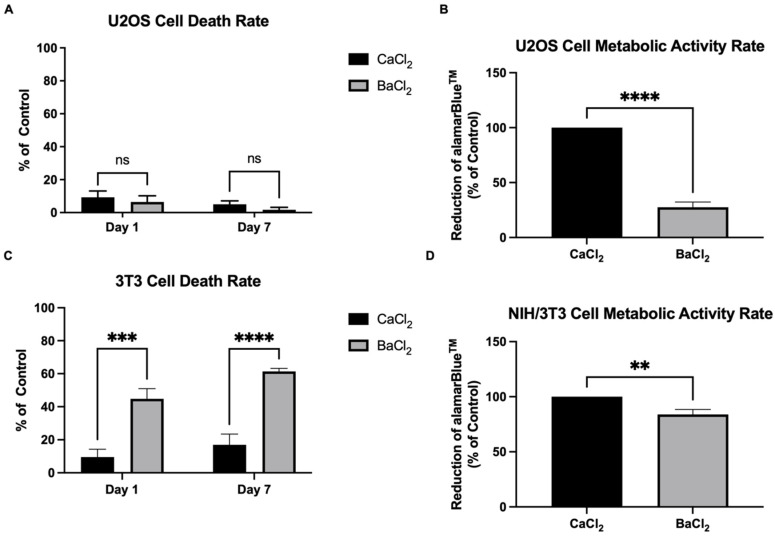
(**A**) LDH cytotoxicity assay of U2OS cells on alginate–gelatin hydrogel at pH 7.0 crosslinked with either 100 mM CaCl_2_ or 100 mM BaCl_2_ on days 1 and 7 of cell culture (day 1 n = 4 and day 7 n = 3, Two-way ANOVA, ns: not significant). (**B**) U2OS metabolic activity rate quantified by the % of alamarBlue^TM^ reduction on day 7 of cell culture on alginate–gelatin hydrogel at pH 7.0 crosslinked with either 100 mM CaCl_2_ or 100 mM BaCl_2_ (n = 4, Two-way ANOVA, **** *p* < 0.0001). (**C**) LDH cytotoxicity assay of NIH/3T3 cells on alginate–gelatin hydrogel at pH 7.0 crosslinked with either 100 mM CaCl_2_ or 100 mM BaCl_2_ on days 1 and 7 of cell culture (n = 3, Two-way ANOVA, *** *p* < 0.001, **** *p* < 0.0001). (**D**) NIH/3T3 metabolic activity rate quantified by the % of alamarBlue^TM^ reduction on day 7 of cell culture on alginate–gelatin hydrogel at pH 7.0 crosslinked with either 100 mM CaCl_2_ or 100 mM BaCl_2_ (n = 4, Two-way ANOVA, ** *p* < 0.01).

**Figure 4 gels-09-00853-f004:**
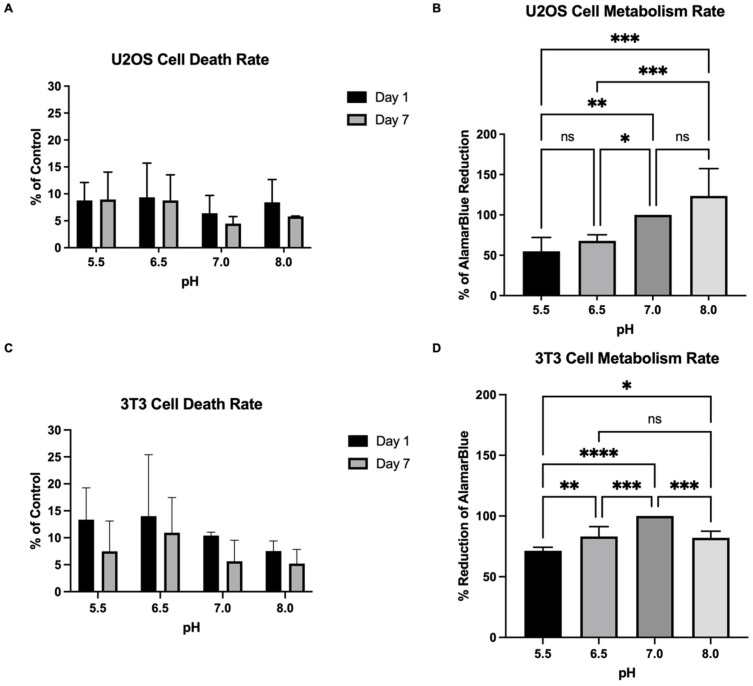
(**A**) LDH cytotoxicity assay of U2OS cells on alginate–gelatin hydrogel at pH 5.5, 6.5, 7.0, and 8.0 on days 1 and 7 of cell culture (day 1 n = 4 and day 7 n = 3, Two-way ANOVA, not significant). (**B**) U2OS metabolic activity rate quantified by the % of alamarBlue^TM^ reduction on day 7 of cell culture on alginate–gelatin hydrogel at pH 5.5, 6.5, 7.0, and 8.0 (n = 6, Two-way ANOVA, * *p* < 0.05, ** *p* < 0.01, *** *p* < 0.001, ns: not significant). (**C**) LDH Cytotoxicity assay of NIH/3T3 cells on alginate–gelatin hydrogel at pH 5.5, 6.5, 7.0, and 8.0 on days 1 and 7 of cell culture (day 1 n = 3 and day 7 n = 3, Two-way ANOVA, not significant). (**D**) NIH/3T3 metabolic activity rate quantified by the % of alamarBlue^TM^ reduction on day 7 of cell culture on alginate–gelatin hydrogel at pH 5.5, 6.5, 7.0, and 8.0 (n = 6, Two-way ANOVA, * *p* < 0.05, ** *p* < 0.01, *** *p* < 0.001, **** *p* < 0.0001, ns: not significant).

**Figure 5 gels-09-00853-f005:**
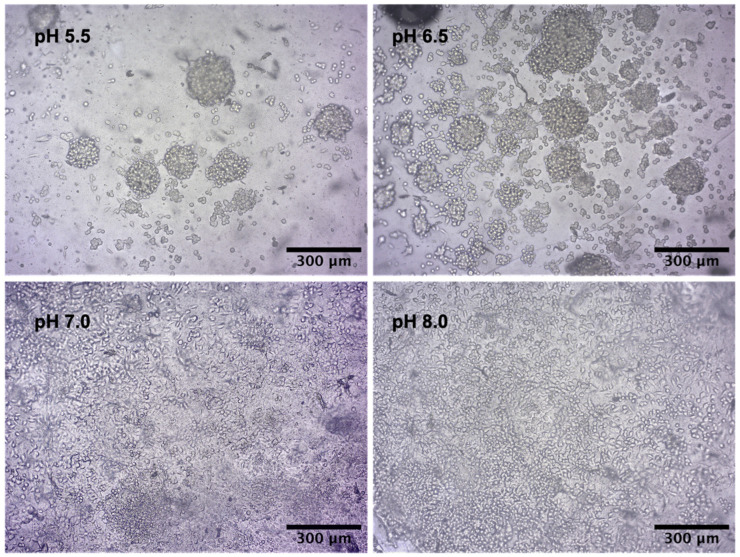
Light microscopy images of U2OS cells cultured on alginate–gelatin hydrogel at pH 5.5, 6.5, 7.0, and 8.0 on day 7 of cell culture. Magnification of 10×.

## Data Availability

The data presented in this study are openly available in article.

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
