# Peer review of "Role of pH and Crosslinking Ions on Cell Viability and Metabolic Activity in Alginate–Gelatin 3D Prints"

_gels, 2023, doi:10.3390/gels9110853_

Round 1
Reviewer 1 Report
In this study, the authors explored how different elements can affect hydrogel stability and printability, and how they influence cell viability and metabolism in resulting 3D prints. This work is interesting, however, before considering acceptance, some problems should be solved.
Here some questions.
What are the different formulations of alginate-gelatin hydrogels used for growing different cell types in vitro?
How do buffer pH and crosslinking ions (Ca2+ or Ba2+) affect the swelling and degradation rates of 3D prints?
What are the implications of this study for bioengineering and tissue regeneration?
The English language was not bad.
Author Response
Thank you very much for your review. We really appreciate your comments and how you attract our attention to important points that should probably be more emphasized in the text. We respond to your questions point by point below. Importantly, the formulations used are typical for the cell types used and similar to those used in our previous studies.
Question 1:
We cited a few references throughout the paper that illustrate the use of alginate-based hydrogels prepared with different formulations, such as different concentrations of alginate and different solvents, for instance. We could add the following papers for better clarification, but we have already reached 49 references.
Di Giuseppe M, Law N, Webb B, Macrae AR, Liew JL, Sercombe BT, Dilley JR, Doyle JB. Mechanical behaviour of alginate-gelatin hydrogels for 3D bioprinting. Journal of the Mechanical Behavior of Biomedical Materials 79 (2018) 150-157. (different concentrations of alginate).
Pan T, Song W, Cao X, Wang Y. 3D bioplotting of gelatin/alginate scaffolds for tissue engineering: influence of crosslinking degree and pore architecture on physicochemical properties. Journal of Materials Science & Technology (2016) 32, 889–900. (used water as a solvent).
Jiang T, Munguia-Lopez GJ, Gu K, Bavoux MM, Flores-Torres S, Kort-Mascort J, Grant J, Vijayakumar S, Leon-Rodriguez DA, Ehrlicher JA, Kinsella MJ. Engineering bioprintable alginate/gelatin composite hydrogels with tunable mechanical and cell adhesive properties to modulate tumor spheroid growth kinetics. Biofabrication (2019) Dec 31;12(1):015024. (used DPBS as a solvent)
Lee YK, Mooney JD. Alginate: properties and biomedical applications. Progress in Polymer Science 37 (2012) 106–126. (Alginate used to grow different cells).
Among other references cited in the paper. Overall, the preparation of alginate-based hydrogels is highly variable and its utilization in the bioengineering field is very broad. If needed, we can add those additional references if the editor allows us beyond the 50 references mark.
Question 2:
We have reported in our paper that both pH and crosslinking have an effect over the hydrogel stability over time under cell culture conditions in the way that the crosslinking with barium increases the alginate stability, slowing down the swelling and degradation rates in comparison to calcium crosslinking. Whereas the pH of the solvent used to produce the hydrogel increases the hydrogel stability, slowing down its swelling and degradation rates, with decreasing pH. pH 8 showed the weakest strength over time under cell culture conditions, leading to increased swelling and degradation rates in comparison to acidic pH values. We have reviewed and edited the manuscript to make sure this message is clear in the Results and Discussion and Conclusion, and we shortened the Abstract.
Question 3:
The implications of our study to the bioengineering field and organ regeneration are that firstly we call the attention of researchers to the effect of the pH and the crosslinking of alginate hydrogels on the hydrogel stability and cell behaviour and how those aspects can lead to different outcomes. Next, we show the effect of the calcium and barium crosslinking as well as the pH on the cell viability and metabolism rate of U2OS cells and NIH/3T3 cells. We stress that substrate pH and what happens to prints in media over time (e.g., swelling, degradation, and ion exchange) is of utmost importance and believe this paper will lead researchers to monitor this aspect more closely and even use it to achieve desired outcomes. Again, we have reviewed and edited the manuscript to make sure this message is clear.
Reviewer 2 Report
This manuscript falls under the Gels scope and presents findings of research on the “Role of pH and Crosslinking Ions on Cell Viability and Metabolic Activity in Alginate-gelatin 3D Prints”. The manuscript consists of 17 pages, 5 figures and 49 literature references. The paper presents interesting results as well as an inquisitive and reliable interpretation of the research results. The topic original and relevant in the field of study. The Abstract provides the highlights of the key contents of the main text. The Introduction provides enough background information to justify the study. The Results are consistent with the declared methodology, presented clearly enough, supported by the figures and tables. Researchers evaluated how the alginate-gelatin pH and the crosslinking ions can affect hydrogel stability and printability and influence U2OS and NIH/3T3 cell viability and metabolism on the resulting 3D prints, and concluded that that both pH and crosslinking ion influence hydrogel strength and cell behavior. The methodology adequately described and conclusion consistent with the evidence and arguments presented. The references are appropriate and relevant to the research. However, minor typographical and grammatical errors need addressing.
Minor typographical and grammatical errors need addressing.
Author Response
Thank you very much for your review. We really appreciate your comments, and we have reviewed the grammar of the paper again and addressed the errors.
Reviewer 3 Report
The manuscript in question is complete in all its parts. I didn't find any critical issues.
The English can be improved and I suggest careful reading.
There are 2 points to improve
1 The abstract is excessive and needs to be shortened. Focus only on results.
2. In my opinion, the authors should better motivate the applications of the results found.
All figures need improvement. I suggest using different colors, different styles for the lines. Furthermore, some fonts may be enlarged.
Improved
Author Response

(The authors gave the same response as above.)

Round 2
Reviewer 1 Report
ACCEPT